# Comparison of Random Regression Models with Different Order Legendre Polynomials for Genetic Parameter Estimation on Race Completion Speed of Arabian Horses

**DOI:** 10.3390/ani12192630

**Published:** 2022-09-30

**Authors:** Hasan Önder, Uğur Şen, Dariusz Piwczyński, Magdalena Kolenda, Magdalena Drewka, Samet Hasan Abacı, Çiğdem Takma

**Affiliations:** 1Department of Animal Science, Ondokuz Mayis University, 55139 Samsun, Turkey; 2Department of Agricultural Biotechnology, Ondokuz Mayis University, 55139 Samsun, Turkey; 3Department of Animal Biotechnology and Genetic, Faculty of Animal Breeding and Biology, Bydgoszcz University of Science and Technology, 85-796 Bydgoszcz, Poland; 4Department of Animal Breeding and Nutrition, Faculty of Animal Breeding and Biology, Bydgoszcz University of Science and Technology, 85-796 Bydgoszcz, Poland; 5Department of Animal Science, Ege University, 35040 İzmir, Turkey

**Keywords:** race completion speed, Arabian horses, heritability, repeated measures, Legendre

## Abstract

**Simple Summary:**

This study aimed to compare different order Legendre polynomials for Arabian racing horses. The present work’s results indicated that the L(2,2) Legendre polynomial (that suggests generalizations and connections to the various mathematical structures and numerical applications) model could be reliably used to estimate heritability values for the racing speed of Arabian horses in the presence of repeated observations.

**Abstract:**

This work aimed to compare the fitting performance of the random regression models applied to the different order orthogonal Legendre polynomials on the race completion speed (m/s) of Arabian racing horses. Legendre polynomial function for additive genetic, permanent environmental variances and heritability values with the L(2,2), L(2,3), L(3,2) and L(3,3) models (where L(i,j) means L(order of fit for additive genetic effects, order of fit for permanent environmental effects)) was estimated. A total of 233,491 race speed records (m/s) of Arabian horses were taken from the Jockey Club of Turkey between 2005 and 2016. The mean and standard deviation of heritability values were estimated as 0.294 ± 0.0746, 0.285 ± 0.0620, 0.302 ± 0.0767 and 0.290 ± 0.1018 for L(2,2), L(2,3), L(3,2), and L(3,3), respectively. The steady decreasing trend of permanent environmental variances for L(2,2) provided stationery for heritability values. According to Akaike information criterion (AIC) and Bayesian information criterion (BIC) values, the L(2,2) model could be reliably used to estimate heritability values for the racing speed of Arabian horses in the presence of repeated observations.

## 1. Introduction

The domestication of horses started 6000 years ago in the southern steppes of Russia [1]. The horse is the animal that played the most memorable and essential role in accelerating the strategic development and social process for the human being [1] with an influence on agriculture, warfare, trade, and transportation [2]. According to 2019 statistics [3], over 59 million horses are bred globally, even though motorized transport has eliminated most reasons for using horses over time. Horses are widely used for entertainment, sports, and recreation. More than 400 diverse breeds have been developed due to selection for many desirable phenotypes [2]. Paleogenomic evidence supports the contribution of ancient Persian ancestry during the early formation of modern European horse breeds about 1100–1300 years ago [4]. One of the most widespread and mainly used breeds is the Arabian horse, used in this study to estimate its genetic parameters. The Arabian horse breed is the oldest recorded horse breed, with reliable documents and pictorial representations dating back 2000 years into the past. Records place the breed’s development in the Middle East [4]. The country of Yemen, Saudi Arabia, Iran, and the Arabian Peninsula is defined as the breed’s place of origin. Still, some previous studies also indicate more remote parts of the Middle East [5]. The breed’s origin is thought to date to distant times, but the history of modern populations of the breed, described in pedigree records, is no longer than 200 years. The first European studs were established in the early nineteenth century in Biała Cerkiew, Sławuta (Poland), Weil (Germany) and Babolna (Hungary), while Arabian breeding in Great Britain and the USA started much later, at the end of the nineteenth century. [5]. Arabian horses originating from the Middle East are one of the world’s oldest breeds. They have been significantly influential in developing new horse breeds such as the “Thoroughbreds” in the modern world [6,7]. Thoroughbreds were developed in England for racing and jumping. The Thoroughbreds’ origins can be traced back to records indicating that a stock of Arab and Barb horses was introduced into England in the early third century [6]. The Arabian horse breed, widely known primarily for its beautiful appearance and endurance, has been used to improve some characteristics of other horse breeds. [7]. Although the Arabian horse breed is still the most common worldwide, with a pedigree recorded in at least 82 countries, the number of American Quarter Horses is now more significant in absolute terms. Arabians are also highly appreciated as racehorses and earn considerable rewards throughout their gallop racing careers [4,8]. In addition, the Arabian horse breed is recognized as the leading breed in endurance competitions [9], but there was no evidence that this breed was genetically superior to other racehorse breeds [10]. Machmoum et al. [11] argued that the genetic diversity of the Arabian horse breed decreases near the risk threshold, which should be considered in breeding studies.

Horse racing can be considered a performance test to evaluate the speed and endurance of horses. In Turkey, the Purebred Arabian horse racing sector size is comparable to that of Thoroughbreds, as the racing merit is a critical element of the breeding goal for Arabian horses. Horse races have been officially held in Turkey since 1913. According to 2019 data, the contribution of horse races to the Turkish economy has been determined as 992,500 USD [12]. The General Directorate of Agricultural Enterprises of the Ministry of Agriculture and Forestry (TİGEM) earned 3 million 54 thousand USD from the sales of 273 Arabian horses in 2021. The average price of a running horse has been reported as 7100 USD [13]. Speed is a fundamental trait to evaluate for horses competing in a race, and it has been widely used in racing horses (the classical “time per km”) [14]. Other common traits are the placing and the annual or career earnings [15]. Elite Arabian racing mares and stallions usually produce elite racing offspring, and this relationship is observed through many generations [16]. Many studies were carried out to estimate the heritability value of the “performance” measured by different traits in many racing breeds, such as Thoroughbred, Arabian, and trotter horses [6,14,15]. However, race speed has never been rated before in Turkey, as well as finish time, position, and earnings.

The selection of breeding animals is of great importance. Generally, the selection of the horses is made by looking at phenotypes such as the number of winnings. Horses that will become parents of the next generation should be selected for the highest possible genetic value. However, it is essential to use phenotypic values and correctly determine the effective factors to estimate the breeding value accurately and reliably [17,18]. It is also known that the statistical method used significantly affects the reliability of the results [19]. The chosen statistical methods also limit the statistical power of the studies because it must ensure that all important environmental factors are included in the analysis and that the experimental error is estimated to a minimum. The random regression models developed for repeated data in recent years are used extensively in determining the breeding values for the economic characteristics of different animals using multiple phenotypic information of an individual [20,21,22]. There is a high interest in predicting the breeding values of animals used in horse racing using random regression models [23]. Legendre polynomials are widely used in random regression models, but their incorrect use (suggesting generalizations and connections to different mathematical structures and numerical applications) in modeling dynamic systems with repeated measures can lead to erroneous results [24,25]. Sufficient work on horse races has not been observed for the fitting performance of different ordered Legendre polynomials, especially on Arabian horses [18]. Therefore, the study aimed to compare the fitting performance of the random regression models applied to the orthogonal Legendre polynomial on the race completion speed (m/s) of Arabian racing horses in this study.

## 2. Materials and Methods

The evaluated data was obtained from the Jockey Club of Turkey between 2005 and 2016 and consisted of 233,491 race speed (m/s) records of 13,764 Arabian horses. The animals with only one race record and/or those without sire and dam information were excluded from the study. Only records up to the tenth race of a horse were used for genetic parameter estimation because the first five races are enough for genetic parameter estimation [6]. After pruning, data of the remaining 12,707 animals were used for the analyses. The number of sires and dams with progeny was 462 and 2274, respectively. The number of animals with offspring was 2736 (21.5%). The number of animals with unknown parents was 2771.

The Legendre polynomial *P_n(x)_* can be defined as for *x* ∈ [−1, 1] and *n* = 1, 2,…,:$$P_{n\left(x\right)}=\frac{1}{2^{n}n!}\frac{d^{n}}{dx^{n}}{\left(x^{2}-1\right)}^{n}$$
where; *d* is the differential operator, *x* is the variable, *n* is the order of the polynomial. The first four Legendre polynomials can be shown as [25]:$$P_{0\left(x\right)}=1$$
$$P_{1\left(x\right)}=x$$
$$P_{2\left(x\right)}=\frac{1}{2}\left(3x^{2}-1\right)$$
$$P_{3\left(x\right)}=\frac{1}{2}\left(5x^{3}-3x\right)$$

L(2,2), L(2,3), L(3,2) and L(3,3) models of Legendre polynomial function, L(i,j) = L(order of fit for additive genetic effects, order of fit for permanent environmental effects), for additive genetic and permanent environmental variances were estimated by the WOMBAT package [26] with the individual animal model method using sire and dam pedigree information. The following mathematical model was used to apply the random regression model:$$y_{ijkl}=RT_{i}+RY_{j}+HA_{k}+\overset{K_{B}}{\underset{m=1}{\sum }}\beta_{m}\left(t_{ij}\right)+\overset{K_{A}}{\underset{m=1}{\sum }}\alpha_{im}\vartheta_{m}\left(t_{ij}\right)+\overset{K_{P}}{\underset{m=1}{\sum }}P_{jm}\vartheta_{m}\left(t_{ij}\right)+e_{ijkl}$$
where; *Y_ijkl_*: the obtained value of race speed (m/s) of the horse *l* at *i*th race track (sand, grass, artificial grass) at *j*th race season (spring, summer, autumn and winter) and the *k*th age (3–12). *RT_i_*: effect of *i*th race tracks, *RY_j_*: effect of *j*th race season, *HA_k_*: effect of *k*th horse age, *β_m_*: *m*th fixed regression coefficients for race number *j*, *t_ij_*: *i*th race of the horse *j*, *α_jm_*: *m*th additive genetic random regression coefficients for horse *j*, *P_jm_*: *m*th permanent environmental random regression coefficients for horse *j*, *φ_m_*: *m*th polynomial evaluated for the race *t_ij_*, *K_B_*, *K_A_* and *K_P_* are the order of fitted fixed, random additive and random permanent regression coefficients, *e_ijkl_*: random residual effect for *y_ijkl_*. The race track distances were not taken as covariates because the speed calculation includes the track distance as a distance divided by the finishing time (m/s). In the study, the race track type, the race season and the horse’s age (in some situations, horses start racing at different ages) were evaluated as fixed effects. The significance of the fixed effects was evaluated using one-way ANOVA, and the means were compared using Duncan’s multiple comparison test.

The −2logL, Bayesian information criterion (BIC), Akaike information criterion (AIC), Log likelihood values and Residual Error Variance (RV) were used to compare the random regression models [18,22,27,28]. The fitting compatibility of the random regression models was also examined regarding the eigenvalues of the covariance matrices [29].

## 3. Results and Discussion

There were statistical differences (*p* < 0.001) for the race speeds obtained in different race tracks type, with the mean and standard error for race speed in different track types being 13.454 ± 0.002, 14.217 ± 0.004 and 14.344 ± 0.002 for sand, synthetic sand, and grass, respectively. All types were found to be statistically different in race speed. There were statistical differences (*p* = 0.029) for race speeds among the seasons for winter, spring, summer, and autumn, with the mean and standard error of 13.819 ± 0.004, 13.837 ± 0.003, 13.831 ± 0.003 and 13.838 ± 0.003, respectively. The race speed was higher for spring than winter and autumn. There was no statistical difference among ages (*p* = 0.160) for race speeds. The descriptive statistics were 13.909 ± 0.003, 13.727 ± 0.003, 13.872 ± 0.004, 13.891 ± 0.005, 13.860 ± 0.005, 13.826 ± 0.007, 13.845 ± 0.009, 13.888 ± 0.022, 13.796 ± 0.014, 13.919 ± 0.019, 13.778 ± 0.020, for ages from 3 to 12, respectively.

Estimates of the additive genetic variance for race speed on different races ranged from 0.077 to 0.327. At the beginning of the race numbers, the additive genetic variance was high but decreased with an increasing number of races for all orders of Legendre polynomials. After the eighth race, all orders of the polynomials except L(3,3) showed a rising trend. The additive genetic variance between the fourth and eighth races had no trend (Figure 1).

Estimates of the permanent environmental variance of race speed for the L(2,2), L(2,3), L(3,2), and L(3,3) Legendre polynomial models changed between 0.046 and 0.156 and trends of the four Legendre polynomial models are illustrated in Figure 2. The permanent environmental variance for L(2,2) and L(3,2) had a decreasing tendency when the number of races increased. Moreover, the Legendre polynomial L(2,3) initially had the highest permanent environmental variance. However, for the last race number, L(3,3) had the highest value among the order of polynomials. L(2,3) and L(3,3) showed wavy, hard-to-interpret graphs. The Legendre polynomials L(2,2) and L(3,2) with descending straight lines were easy to interpret according to animal breeding approaches.

Estimates of the phenotypic variances of race speed for the four Legendre polynomial models have changed between 0.355 and 0.619, and the trends are presented in Figure 3. The phenotypic variances for all four Legendre polynomial models had a decreasing trend. The model obtained from L(2,2) decreased until the seventh race, and the others decreased until race number eight, then all tended to increase.

Estimates of the heritability values of race speed for the four Legendre polynomial models have changed between 0.208 and 0.529, and the trends are presented in Figure 4. Heritability values dramatically decreased until races four and five. The model obtained from L(3,3) showed a wavy graph. The models obtained from L(2,2) and L(3,2) had an increasing trend after race seven.

The model fitting statistics used to compare the models for the L(2,2), L(2,3), L(3,2), and L(3,3) Legendre polynomials are given in Table 1, where the logarithm of the likelihood function, Akaike’s information criterion, Bayesian information criterion and residual variance are illustrated. AIC and BIC values increased with the increasing order of Legendre polynomial, meaning a worse fitting. This inference was not valid for residual error variance.

The maximum log-likelihood (LogL) values and the LogL changes of models with different Legendre polynomial orders are shown in Table 2. The LogL values changed between 7806.1 and 8781.2. The highest change was observed for L(2,3), and no change was observed for L(2,2) and L(3,2).

Eigenvalues of the additive genetic (co)variance matrix and the proportion of total variance (%) estimated from the Legendre models showed that the first eigenvalues could explain over 70% of the total variance for all Legendre polynomials (Table 3).

Eigenvalues of the permanent environmental (co)variance matrix and the proportion of total variance (%) estimated from the Legendre models showed that the first eigenvalues could explain over 80% of the total variance for L(2,2), L(2,3), L(3,2), and L(3,3). For L(2,2), the total variance by the first eigenvalue reached 99.19% as the highest one, and the lowest first eigenvalue explanation was observed for L(2,3) at 83.85% (Table 4).

Many different statistical approaches can be used for modeling repeated measurements in animal science. These models aimed to explain how the traits change over time. The relationships between the results of race numbers are the most important point in repeated measurements. The (co)variance structure between test days (race numbers) is essential to analyze repeated measured data sets.

Random regression analyses with Legendre polynomials of repeated data can describe the genetic variability of results at different time points [30]. The order of the Legendre polynomials has specific importance for the reliability of random regression analysis aimed at solving the linear complexity viewpoint [31]. The comparison of the fit performances of the random regression models applied to the orthogonal Legendre polynomial.

Estimated additive genetic variances for race speed from different order Legendre polynomial models (L(2,2), L(2,3), L(3,2), and L(3,3)) began with higher values and decreased until race numbers five and six. The maximum additive genetic variance was observed for L(3,3) with the value of 0.327 in the first race, and the minimum (0.077) was observed for L(2,3) in the seventh race. After race number eight, an increasing trend was observed for L(2,2) and L(2,3). Önder et al. [23] estimated the additive genetic variance as in inverse S curve for growth traits that differ from current findings. The proportion of total variance (%) calculated from the Legendre models for L(2,2), L(2,3), L(3,2), and L(3,3) for additive genetic (co)variance matrix showed that the first eigenvalues could explain over 70% of the total variance for all models. The explained proportions of the total variance of L(2,3) and L(2,2) were found as 75.68% and 73.64%, respectively. The explained proportion of total variance of the first eigenvalues was not higher than 90%, which does not explain the total variance itself [28]. The best Legendre polynomial model was determined as L(2,3) for the estimated additive genetic effect (Figure 1).

The estimated permanent environmental variances for race speed from different order Legendre polynomial models showed different curve shapes. Legendre polynomial models L(2,2) and L(3,2) showed descending straight lines according to race number, which can be easily interpreted as increasing repeated observations and decreasing permanent environmental variance. Legendre polynomial model L(3,3) showed a sine wave curve, and L(2,3) showed a polynomial. Önder et al. [23] estimated permanent environmental variances as S-shaped and linear curves for Sanen kids’ growth traits. Owaga and Satoh [32] found a polynomial graph for permanent environmental variances, similar to our findings on L(2,3). The proportion of total variance explained from the Legendre models for L(2,2), L(2,3), L(3,2), and L(3,3) for the permanent environmental (co)variance matrix showed that the proportion of total variance of the first eigenvalues could explain over 80% of the total variance for all models. The explained proportions of the total variance of L(2,2) and L(3,2) were found as 99.19% and 97.56%, respectively. Except for L(2,3), the models resulted in a higher than 90% first eigenvalue explanation that can explain the total variance. The best Legendre polynomial model was determined as L(2,2) for the estimated permanent environmental variance (Figure 2).

The estimated phenotypic variances for race speed from different order Legendre polynomial models showed similar curve shapes. All Legendre polynomial models had a decreasing trend until race number seven. After race number eight, model L(2,2) showed an increasing trend while the others continued to decrease. Owaga and Satoh [32] found a polynomial graph for phenotypic variances that did not fit our results (Figure 3). The shape of the estimated phenotypic variances and estimated additive genetic variances were similar. Still, the shape of the estimated permanent environmental variances was different, especially for L(2,3) and L(3,3), which could significantly affect heritability estimates. The findings of Abacı [26], and Owaga and Satoh [32] on the shape of graphs differed from our findings, which the data structure could explain. Further, the shape of the estimated phenotypic variances and additive genetic variances found by Buxadera et al. [24] differed from our findings.

Heritability estimates for race speed from different order Legendre polynomial models for L(2,2), L(2,3), L(3,2), and L(3,3) began with higher values and decreased until race numbers five and six. The maximum heritability was observed for L(3,3) with a value of 0.529 in the first race and the minimum (0.208) for L(3,3) in the tenth race. For the L(2,2), linear decreasing was observed until race number six, and after race number seven, it started to increase, which was the same for L(2,3). Decreasing heritability value for L(3,2) was observed until race number four. The polynomial graph was observed for L(3,3) (Figure 4). These findings differed from Abacı et al. [19], where all Legendre polynomial models showed a static heritability graph for a race number for Thoroughbred racing horses. Furthermore, our findings did not fit the Coskun et al. [6] study on Thoroughbred racing horses. These differences may be caused by their response variable (finishing time). Abacı [28] found linear and quadratic increasing heritability values for Holstein Friesian dairy cattle milk yield. Owaga and Satoh [32] found a cubic Bezier-type graph for heritability values for the calving interval of Japanese Black cows. Gómez et al. [33] found linear type increasing heritability values for annual earnings of Trotter horses. The heritability values obtained from other studies were between 0.2 and 0.3, and in our results, the mean and standard deviation of heritability values were 0.294 ± 0.075, 0.285 ± 0.062, 0.302 ± 0.077 and 0.290 ± 0.102 for L(2,2), L(2,3), L(3,2), and L(3,3), respectively. Coefficient of variation (CV) values of heritability estimates were 25.39, 21.78, 25.45 and 35.18 for L(2,2), L(2,3), L(3,2), and L(3,3), respectively.

According to the interpretation of all graphs, waving of permanent environmental variances affected the heritability estimates for L(3,3), which were not robust due to a CV value over 0.30. For L(2,3), the polynomial curve of permanent environmental variances did not cause a waving of heritability value but decreased it. The stability of permanent environmental variances for L(3,2) increased the heritability estimates. The steady decreasing graph of permanent environmental variances for L(2,2) provided stationery for heritability values. When the proportion of the total variance of the additive genetic (co)variance matrix and the permanent environmental (co)variance matrix were evaluated together with the identification of AIC and BIC criteria, the L(2,2) Legendre polynomial model was shown to be the best model to estimate the heritability values in comparison with L(2,3), L(3,2) and (L3,3). Using the L(2,2) polynomial model to estimate genetic parameters for Arabian horses may give more reliable results than other investigated models.

Furthermore, it should be remarked that the breeding programs are generally focused on the improvement of riding traits. This focus causes intensive selection, which usually leads to a reduction in genetic variability and a gradual increase in inbreeding. The inbreeding effect was dramatically increased for traits such as wither height, decreasing heritability. In addition, new methods, such as marker-assisted and genomic selection for horse breeding can be used for genetic parameter estimation for racing horses.

## 4. Conclusions

Many researchers use L(3,2) or L(3,3) Legendre polynomial models, which are generally used in dairy milk yield studies that are not fit for racing horse studies. The results of the present work indicated that the L(2,2) Legendre polynomial model could be reliably used to estimate heritability values for the racing speed of Arabian horses in the presence of repeated observations. Thus, creating mating programs according to this information, by using the Legendre polynomial (2,2) to estimate genetic parameters, especially breeding value, could improve the breeding achievements in Turkey.

## Figures and Tables

**Figure 1 animals-12-02630-f001:**
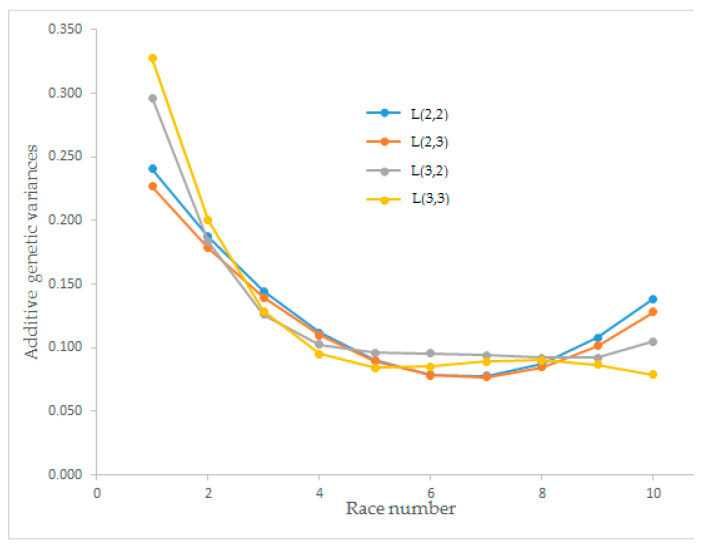
Estimation of the additive genetic variance for race speed from different orders of Legendre polynomial models.

**Figure 2 animals-12-02630-f002:**
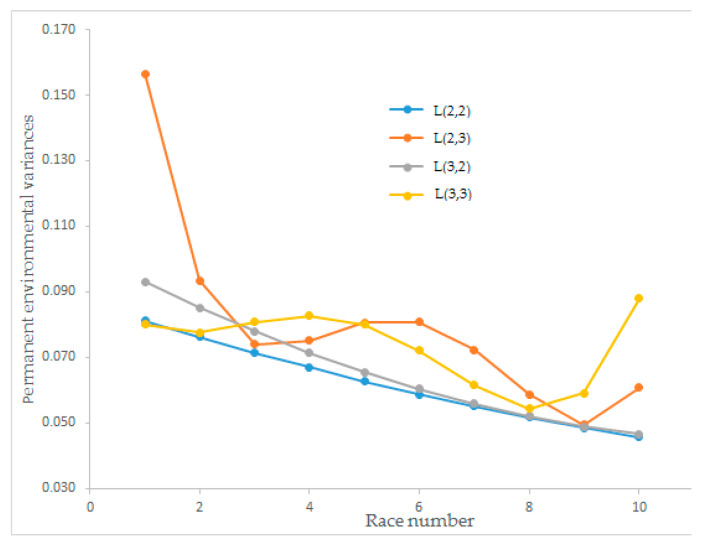
Estimation of permanent environmental variance for race speed from different orders of Legendre polynomial models.

**Figure 3 animals-12-02630-f003:**
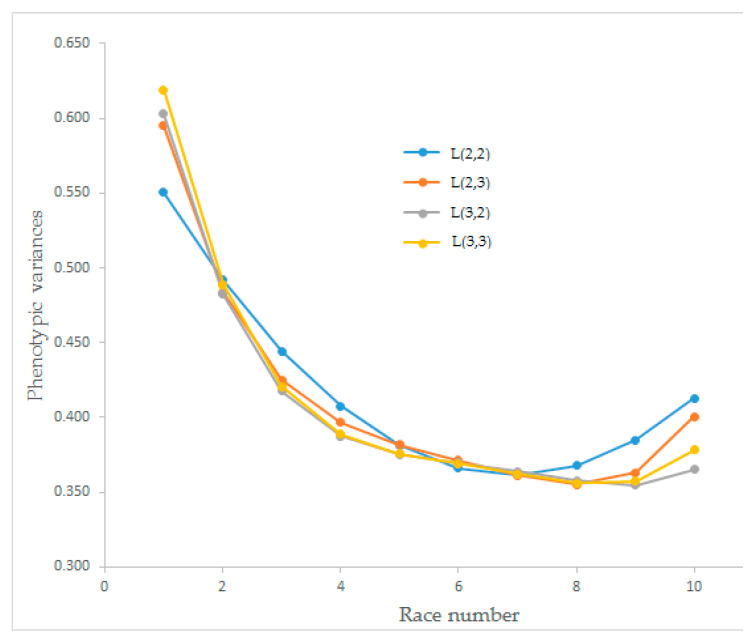
Estimation of the phenotypic variances for race speed from different orders of Legendre polynomial models.

**Figure 4 animals-12-02630-f004:**
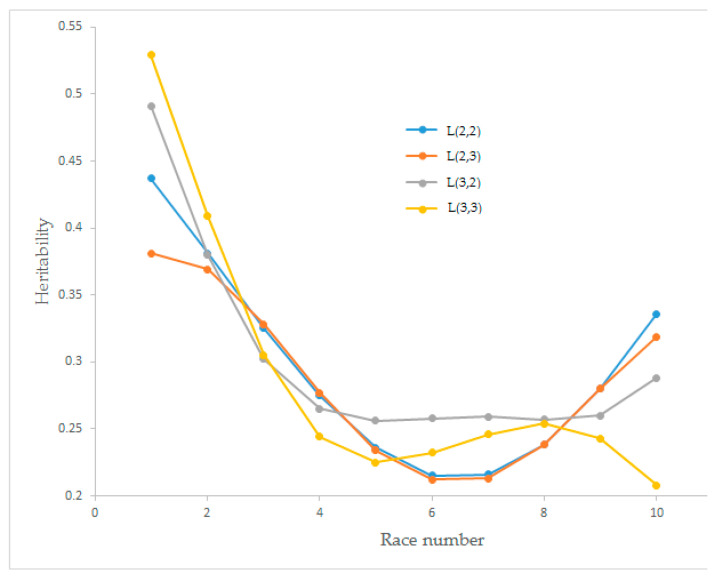
Estimation of heritability values for race speed from different orders of Legendre polynomial models.

**Table 1 animals-12-02630-t001:** Criteria used for comparison of the models.

Models	Number of Parameters	Maximum LogL	AIC	BIC	RV
L(2,2)	7	7806.1	7798.1	7760.9	0.228
L(2,3)	11	8342.2	8331.2	8280.0	0.212
L(3,2)	11	8781.2	8770.2	8718.9	0.214
L(3,3)	14	8852.2	8838.2	8773.0	0.211

**Table 2 animals-12-02630-t002:** Maximum log-likelihood values and changes in the log likelihoods at the different models.

Models	Number of Parameters	Log Likelihood	Changes in Log Likelihood	Changes in Log Likelihood (%)	χ^2^
L(2,2)	7	7806.1	-	-	-
L(2,3)	11	8342.2	+536.1 *	6.42	7.81
L(3,2)	11	8781.2	-	-	-
L(3,3)	14	8852.2	+71 *	0.80	7.81

*: Significant change (*p* < 0.05).

**Table 3 animals-12-02630-t003:** Eigenvalues of the additive genetic (co)variance matrix and the proportion of total variance (%) estimated from the Legendre models.

Models	First	Second	Third
L(2,2)	0.17 (73.64)	0.06 (26.36)	-
L(2,3)	0.17 (75.68)	0.06 (24.32)	-
L(3,2)	0.17 (71.06)	0.06 (24.87)	0.01 (4.06)
L(3,3)	0.17 (72.23)	0.06 (27.40)	0.00 (0.37)

**Table 4 animals-12-02630-t004:** Eigenvalues of the permanent environmental (co)variance matrix and the proportion of total variance (%) estimated from the Legendre models.

Models	First	Second	Third
L(2,2)	0.12 (99.19)	0.00 (0.81)	-
L(2,3)	0.13 (83.85)	0.02 (15.61)	0.00 (0.54)
L(3,2)	0.13 (97.56)	0.00 (2.44)	-
L(3,3)	0.13 (90.07)	0.01 (9.47)	0.00 (0.46)

## Data Availability

To obtain the data, please contact the author H.Ö.

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
