# Peer review of "Comparison of Random Regression Models with Different Order Legendre Polynomials for Genetic Parameter Estimation on Race Completion Speed of Arabian Horses"

_animals, 2022, doi:10.3390/ani12192630_

Round 1

Reviewer 1 Report (New Reviewer)

Introductions

L(2,2), L(2,3), L(3,2) and L(3,3) models of Legendre polynomial

function for additive genetic, permanent environmental variances and heritability values were estimated

Please specify before what L(2,2), L(2,3), L(3,2), and L(3,3) mean.

233491 race speed (m/sec) records of 13764 Arabian horse raced taken from the Jockey Club of Turkey between 2005 and 2016 were used

You cannot start a sentence with a number

0.2939±0.0746

Three digits are enough

One of the important breeds in those is the Arabian horse which is the issue of this study

“Issue” does not suit properly, better study subject

The origin of the breed is thought to date back to ancient times, but modern populations described in pedigree records are no longer than 200 years old

I would improve that phrase

Thoroughbred is developed in England for racing and jumping.

Thoroughbred was developed in England for racing and jumping.

In the light of these data, it can be said that Arabian horse breeding and races have an important role in the Turkish economy

I know that makes sense, but I don’t consider that sentence can fit in a scientific article without a citation.

But racing speed didn’t evaluate before in Turkey conditions even finishing time, finishing position and earnings were evaluated

However, race speed has never been rated before in Turkey, as well as finish time, finish position, and earnings.

However, in order to estimate the genotypic value reliably, phenotypic values should be used, and effective factors should be determined correctly [17,18]. In addition, the chosen statistical method has a significant effect on the reliability of the result [19]. The statistical power of the studies has also been restricted by the statistical methods and computational resources that make it difficult to include all significant environmental factors in the analyses and potentially biasing estimate

Please, rewrite this sentence

To predict genetic parameters

To estimated

To predict genetic parameters, there are many types of methods such as the sire model or animal model within the best linear unbiased prediction (BLUP), restricted maximum likelihood (REML), random regression (RR), and Bayesian methods 

This sentence is absolutely wrong, it should be eliminated

Materials and Methods

Only records up to the 10th race of a horse were used for genetic parameter estimation

Why that?

13.45 ± 0.002, 14.22 ± 0.004 and 14.34 ± 0.002

For a matter of consistency, I would suggest using 3 digits

In the study, the race track type, the race season and the age of the horse (in some

situations horses starts to racing in different ages) were evaluated as fixed effects. There

was statistically differences (P<0.001) among race tracks type with mean and standard

error for race tracks type were 13.45 ± 0.002, 14.22 ± 0.004 and 14.34 ± 0.002 for sand, synthetic sand and grass, respectively. All types were found statistically different on speed.

There was statistically differences (P=0.029) among seasons for winter, spring, summer

and autumn with the mean and standard error of 13.83 ± 0.004, 13.84 ± 0.003, 13.83 ± 0.003

and 13.84 ± 0.003, respectively. The speed was higher for spring than winter and autumn.

There were no statistical difference among ages (P=0.160). The descriptive statistics were

13.91 ± 0.003, 13.73 ± 0.003, 13.87 ± 0.004, 13.89 ± 0.005, 13.86 ± 0.005, 13.83 ± 0.007, 13.84 ±

0.009, 13.89 ± 0.022, 13.80 ± 0.014, 13.92 ± 0.019, 13.80 ± 0.020, for ages from 3 to 12, respectively

That part is interesting however I would suggest putting on results and just briefly mentioned on M&M

Owaga and Satoh [32] found circular curve for additive genetic variance for calving interval of Japanese Black cows which was too different from our findings

I would remove that.

Author Response

For Reviewer 1

Reviewer Comment

L(2,2), L(2,3), L(3,2) and L(3,3) models of Legendre polynomial function for additive genetic, permanent environmental variances and heritability values were estimated

Please specify before what L(2,2), L(2,3), L(3,2), and L(3,3) mean.

Answer

The description of the Legendre polynomial functions were given before the phrase “L(2,2), L(2,3), L(3,2) and L(3,3) models of Legendre polynomial function for additive genetic, permanent environmental variances and heritability values were estimated”. Please see lines 138-148 in the text.

Reviewer Comment

233491 race speed (m/sec) records of 13764 Arabian horse raced taken from the Jockey Club of Turkey between 2005 and 2016 were used

You cannot start a sentence with a number

Answer

The sentence was reorganized as “The evaluated data was obtained from the Jockey Club of Turkey between 2005 and 2016 and consisted of the 233491 race speed (m/sec) records of 13764 Arabian horses.” Please see lines 127-128 in the text.

Reviewer Comment

0.2939±0.0746

Three digits are enough

Answer

Writing about the digits were corrected. Please see line 449 in the text.

Also, belong to this correction that numbers were corrected in Abstract section. Please see line 28 in the text.

Reviewer Comment

One of the important breeds in those is the Arabian horse which is the issue of this study

“Issue” does not suit properly, better study subject

Answer

The sentence was corrected as “One of the important breeds in those is the Arabian horse which is aimed in this study to estimate its genetic parameters.” Please see lines 45-46 in the text.

Reviewer Comment

The origin of the breed is thought to date back to ancient times, but modern populations described in pedigree records are no longer than 200 years old

I would improve that phrase

Answer

The phrase was improved as “The origin of the breed is thought to date to distant times, but the history of modern populations of the breed, described in pedigree records, is no longer than 200 years. The first European studs in Biała Cerkiew, Sławuta (Poland), Weil (Germany) and Babolna (Hungary) were established in the early nineteenth century while Arabian breeding in Great Britain and the USA started much later at the end of nineteenth century.” Please see lines 52-57 in the text.

Reviewer Comment

Thoroughbred is developed in England for racing and jumping.

Thoroughbred was developed in England for racing and jumping.

Answer

The change was done is à was.  Please see line 75 in the text.

Reviewer Comment

In the light of these data, it can be said that Arabian horse breeding and races have an important role in the Turkish economy

I know that makes sense, but I don’t consider that sentence can fit in a scientific article without a citation.

Answer

This sentence was deleted. Please see lines 84-86 in the text.

Reviewer Comment

But racing speed didn’t evaluate before in Turkey conditions even finishing time, finishing position and earnings were evaluated

However, race speed has never been rated before in Turkey, as well as finish time, finish position, and earnings.

Answer

The sentence was replaced with “However, race speed has never been rated before in Turkey, as well as finish time, finish position, and earnings.” Please see lines 92-94 in the text.

Reviewer Comment

However, in order to estimate the genotypic value reliably, phenotypic values should be used, and effective factors should be determined correctly [17,18]. In addition, the chosen statistical method has a significant effect on the reliability of the result [19]. The statistical power of the studies has also been restricted by the statistical methods and computational resources that make it difficult to include all significant environmental factors in the analyses and potentially biasing estimate

Please, rewrite this sentence

Answer

The sentences were rewritten as “However, it is important to use phenotypic values and to determine the effective factors correctly in order to estimate the breeding value accurately and reliably [17,18]. It is also known that the statistical method used has a significant effect on the reliability of the results [19]. The statistical power of the studies is also limited by the statistical methods chosen because it must be ensured that all important environmental factors are included in the analysis and that the experimental error is estimated to a minimum.” Please see lines 99-105 in the text.

Reviewer Comment

To predict genetic parameters

To estimated

Answer

This sentence was deleted according to next comment. Please see line 105 in the text.

Reviewer Comment

To predict genetic parameters, there are many types of methods such as the sire model or animal model within the best linear unbiased prediction (BLUP), restricted maximum likelihood (REML), random regression (RR), and Bayesian methods

This sentence is absolutely wrong, it should be eliminated

Answer

This sentence was deleted. Please see lines 105-113 in the text.

Reviewer Comment

Only records up to the 10th race of a horse were used for genetic parameter estimation

Why that?

Answer

The sentence was improved as “Only records up to the 10th race of a horse were used for genetic parameter estimation be-cause the first five races are enough to genetic parameter estimation [6].” Please see line 133 in the text.

Reviewer Comment

13.45 ± 0.002, 14.22 ± 0.004 and 14.34 ± 0.002

For a matter of consistency, I would suggest using 3 digits

Answer

The corrections were done. Please see lines 197-205 in the text.

Reviewer Comment

In the study, the race track type, the race season and the age of the horse (in some situations horses starts to racing in different ages) were evaluated as fixed effects. There was statistically differences (P<0.001) among race tracks type with mean and standard error for race tracks type were 13.45 ± 0.002, 14.22 ± 0.004 and 14.34 ± 0.002 for sand, synthetic sand and grass, respectively. All types were found statistically different on speed. There was statistically differences (P=0.029) among seasons for winter, spring, summer and autumn with the mean and standard error of 13.83 ± 0.004, 13.84 ± 0.003, 13.83 ± 0.003 and 13.84 ± 0.003, respectively. The speed was higher for spring than winter and autumn. There were no statistical difference among ages (P=0.160). The descriptive statistics were 13.91 ± 0.003, 13.73 ± 0.003, 13.87 ± 0.004, 13.89 ± 0.005, 13.86 ± 0.005, 13.83 ± 0.007, 13.84 ±0.009, 13.89 ± 0.022, 13.80 ± 0.014, 13.92 ± 0.019, 13.80 ± 0.020, for ages from 3 to 12, respectively

That part is interesting however I would suggest putting on results and just briefly mentioned on M&M

Answer

This part moved to beginning of the Results section. Please see line 194-205 in the text.

Reviewer Comment

Owaga and Satoh [32] found circular curve for additive genetic variance for calving interval of Japanese Black cows which was too different from our findings

I would remove that.

Answer

This sentence was deleted. Please see line 387-389 in the text.

Reviewer 2 Report (New Reviewer)

This is an interesting paper which was aimed to compare the fitting performance of the random regression models applied to the different order orthogonal Legendre polynomials on some physiological parameters in Arabian racing horses. The reviever became acquianted with the comment of the second (third ?)  previous reviewer which greatly simplifies the task. In my opinion the manuscript in the present form with all corrections made during the revision  process can be published in the present form.

Author Response

For Reviewer 2

Reviewer Comment

This is an interesting paper which was aimed to compare the fitting performance of the random regression models applied to the different order orthogonal Legendre polynomials on some physiological parameters in Arabian racing horses. The reviever became acquianted with the comment of the second (third ?)  previous reviewer which greatly simplifies the task. In my opinion the manuscript in the present form with all corrections made during the revision  process can be published in the present form.

Answer

All the authors thanks to reviewers valuable contributions that made this manuscript more attractive.

Reviewer 3 Report (New Reviewer)

The authors estimated the genetic parameters of race completion speed by several methods using 233491 race speed records of 13764 Arabian horses. Overall, the statistical analyses were conducted correctly and the results make sense. Below please find my comments.

1. Why classical methods for estimating genetic variances, such as AI-REML or EM-REML, were not performed to compare with the recommended Random Regression Models?

2. The abbreviations in all figures and tables, e.g., 2-2, L(2,2), should be clearly explained in the legends.

Author Response

For Reviewer 3

Reviewer Comment

Why classical methods for estimating genetic variances, such as AI-REML or EM-REML, were not performed to compare with the recommended Random Regression Models?

Answer

In the text (Lines 114-116) it was mentioned that “the random regression models developed for repeated data in recent years are used extensively in determining the breeding values for the economic characteristics of different animals”

We used Random regression to taken into consideration the repeated races.

Reviewer Comment

The abbreviations in all figures and tables, e.g., 2-2, L(2,2), should be clearly explained in the legends.

Answer

All tables and Figures were corrected. Please see line 213, 251, 277, 306, 333, 348, 356, and 367 in the text.

Round 2

Reviewer 1 Report (New Reviewer)

The manuscript in the present form can be published now 

Author Response

Thank you for your valuable effort

This manuscript is a resubmission of an earlier submission. The following is a list of the peer review reports and author responses from that submission.

Round 1

Reviewer 1 Report

I believe that the authors have an excellent data set and manifest the required level for the analysis of them, however many details have been lacking.

The introduction provides an explanation of the evolution of horses but does not provide any details on the importance of this breed of horses and that activity for his own country. Authors should not be so categorical in mentioning (l 77) that there is no published evidence on speed, when the literature exists. The objectives must be specified for the Arabian horse in Turky conditions.

In material and method, they must incorporate the information of how many ancestors were represented in the animal vector. It is not correct to assume that the 'speed' character is the same at different distances (l 112), there is evidence available to prove that this is not correct Before presenting components of (co)variance they must shows some descriptive statistics. Finally, the proposed model is the best with respect to which? What is the contribution of your model respect to what another model? How can your model help the breeding program of this breed in your country?

Author Response

the authors thank reviewers for their valuable comments that improve the quality of this manuscript.

Reviewer 2 Report

Önder and colleagues report a study on different regression models to investigate phenotypic and genetic traits of Arabian horses.

The manuscript deals with an original topic. Nevertheless, the manuscript should be improved.

The manuscript is not suitable for publication in the present version.

The comments I report, both general and specific, are intended for a constructive revision.

One of the main weaknesses derives from insufficient editing of English language and punctuation. I report some specific comments but the manuscript should be entirely examined.

I am sure the authors can improve the manuscript.

This and other issues reported in the list could improve the manuscript and must be addressed before publication.

Specific comments:

Lines 17-20: the simple summary, according to instructions to authors, should “be written for a lay audience, i.e., no technical terms without explanations.” Please revise.

Lines 21-30: the abstract must be improved for the reader. For example, in lines 26-28 the authors report that “The maximum heritability was observed for L(3,3) model with the value of 0.529 at first test day and the minimum (0.208) was observed for L(3,3) model at tenth test day.”; and in lines 28-30 that “In conclusion, the results of the present work indicated that L(2,2) model can be reliably used to estimate…”. What’s the relationship between the two models and the two sentences? The abstract should be self-standing.

Lines 77-78: “But racing speed didn’t evaluated before.” This sentence is not clear.

Lines 94-97: a short introduction of the Legendre polynomials could improve this part.

Lines 104-106: “Records up to the 10th run of a horse have been used. The animals were recorded less than two races, and the animals that parents couldn’t reach were excluded from the study. After pruning” This sentence is not clear. Also, what’s the approach for data editing and analyse only 12707 out of 13764 animals?

Line 113: the description of the differences among the L(2,2), L(2,3), L(3,2) and L(3,3) models could improve this part for the reader.

Line 138, 139 and following: “After the number of races at eight”. “…between four and eight number of races…”. These sentences (and the following of similar type) and description of races’ numeration should be improved. I also suggest presenting the concept that race number is equivalent to test day (e.g. at line 301 “After test day eight..”). The presentation of data and discussion would be improved for the reader.

Lines 378-388: citations in the paragraph Conclusions are not forbidden, but unusual. Some concepts could be moved to the paragraph Discussion.

Author Response

(The authors gave the same response as above.)

Reviewer 3 Report

The authors fit random regression models for the speed trait in horse races. Random regression models are fitted when the genetic performance can vary across a covariate, existing  animals that can perform better in some points of the covariate and others that perform better in other points. Surprisingly authors do not refer to any covariate performance-dependant in the introduction, so that the reader cannot identify which is the environmental factor to be fitted. It looks as if the authors only worried about fitting a random regression model without any base consistent idea, just mimicking what other researchers have done for number of lactations of test day models. I can see later in a figure that the covariate fitted is called "Test Days", but I cannot find any definition for this. From the editing (L114-116) I can infer that the covariate refers to the number of races ran from 1 to 10 when available. On one hand, as there is nothing commented about this in the introduction section, the research remains unjustified. Secondly, the definition of "Test Days" looks completely out of order: it seems that the authors simply copied what other authors reported analyzing milk production according to the lactation curve, but it has no sense here. And finally, even when admitting that different horses can differently change their performance depending on the number of races ran, I cannot see the limitation to the 10th race. I am not an expert in horses but I understand that many horses will run 10 or 20 races each year, totaling more than 100 across their life. Then the study has been restricted for the most early sportive life of the horse, which is probably not interesting. Why not use all the races ran for the animal performing the more? Why not alternatively use the age of the horse as the covariate to fit the random regression? Summarizing I suspect that the authors do not know what they have done, and it is reinforced by the inconsistencies appearing in the model definition and the explanations in the results and discussion sections.

Another point, the authors do not use random regression models to find horses with different behaviour in different number of races. They focused on the best model across the four they essayed. If the objective is this, can the authors assess that random regression models are better than the traditional models assuming a unique variance component across number of races? There is no comparison at all with that classical model as reference.

Discussion section gathers all the concerns reported above, as much of the discussion is a repetition of  the results. Authors deal with the shape of the plots, but this has null transcendence for the genetic evaluation, the breeding scheme or whatever. The comparisons the authors do are with other livestock populations completely incomparable with the performance f horses across their first 10 participations in races.

Summarizing, the utility of this research is unknown, the implications in the breeding process do not exist or have not been presented, the covariate used to define as random regression is doubtful to be appropriate, the model lacks of rigor, the genetic language is inadequate, and the results are poorly discussed. I find the manuscript rejectable in its present form. It looks that the data set is interesting and can be worthy to analyze, but considerable improvement is needed yet, and I have strong doubts the authors can achieve it. I leave the editors to decide about that.

I am not a native English speaker but even so I can realize that the English grammar and style needs improvement.

Detailed revision:

L19-20 I disagree about the pertinence of the added comment.

L28 I guess "363" is a mistake and has to be erased.

L53-56 I do not know the history of the different breeds, but I find completely different today an Arab and a Thouroughbred animal. I guess that the genetic connection between them falls some centuries ago. I would not remark this connection.

L98-100 Here all is completely mixed. BLUP and REML are methods, animal o sire models are models within methods, random regression is a type of effect within models and within methods, bayesian is almost a different paradigm with regards to frequentist classical statistics.

L105-106 Again I disagree with adding this comment.

L114 What is test day race speed? It is very clearly defined what test day milk yield is in the scope of the dairy production, but it is not clear what it means here.

L117-118 I cannot see a reason for these restrictions, and this can lead to biased sampling. On the other hand keeping animals without pedigree information is useful for a correct estimation of the trajectory of the performances across number of races.

L122 I am not sure that the age effect has to be included in the model because I can understand there must be colineality between the age and the test day.

L127-128 The number of decimal figures must match between the means and the standard error because as it is now, it is impossible to assess which of the means are statistically different. On the other hand, probably the definition of the levels of the season effect does not have to be done according to the calendar. Sometimes the environmental influence does not change in the point in which the seasons are defined, and seasonality has to be defined according to the performance.

L130-132 Again, use the same decimal figures for means and standard errors.

L132-134 Disagree. It is obvious that mean speed will trend to decrease when the distance is longer. It is because of that that there are breeds specialized for short and for long distances, being clearly higher the average speed in shorter distances.

L134-135 Polynomials are defined based on a continuous covariate. I guess it is the number of race of the horse but it is ignored. Authors omit the covariate, and it is too important to do that.

L143 I wonder why the age effect was fitted. On one hand authors report no difference between age classes (L129-132), and on the other hand, this effect looks colinear with which what I suppose is the covariate defined for the random regression.

L144-145 I think this is not well defined. Betam is the fixed regression for the ¿"race number"?, not for the horse.

L145-147 Additive genetic and permanent effects should not have to be identified with the same j subindex, as the first one must include animals in the pedigree without information, while the second one only refers to animals with records.

L147 KB, KA and KP have to be specified that go from 2 to 3.

L154-163 I find these added lines inappropriate. In case of keeping them, all the terms there have to be defined. For example, what is d?

L165-170 Here and across the text, these are not the estimates of the additive genetic effect. It could probably be the estimate of the additive genetic variance. I mean, what you are plotting is the additive genetic variance across number of races ran by a horse.

L199-207 and others... Again, this is not the permanent effect, but the permanent environmental variance.

Figure 1 and 2. Captions and legends have to be modified according to all I concerned in this review.

L281 "are given"

L298-315 I do not see the usefulness of the eigenvalues analyses here. Can the authors explain it?

L405-407 This is only comparing among the four random regression models essayed here.

L410-412. Only L(2,2) is more reliable than the other three, but assessing that it is reliable is not shown in this manuscript.

L412-414 This has not been demonstrated in this paper.

L415-416 I think that comparing this analysis with milk populations has led the authors to unfocus their research.

L416-423 This is not concluded from the analyses done here. Maybe this can be moved to the introduction section.

Round 2

Reviewer 1 Report

The authors have not considered the fundamental criticism made and reiterate that there is no evidence on racing speed (l 86-89) which is not correct. I copy a few references:

Bülent EKZ, Ömür KOÇAK, and  Hedir Demir, 2005. Estimates of Genetic Parameters for Racing Performances of Arabian Horses. Turk J Vet Anim Sci 29 543-549

Mota, M.D.S., Oliveria, H.N., Silva, R.G,1009. Genetic and environmental factors that affect the best time of Thoroughbred horses in Brazil. J. Anim. Breed. Genet.,  115: 123-129.

Gomez M.D., A. Menendez-Buxadera , M. Valera2 & A. Molina,2010. Estimation of genetic parameters for racing speed at different distances in young and adult Spanish Trotter horses using the random regression model .  J. Anim. Breed. Genet. 127:  385–394

 Oki H., Y. Sasaki and ,R.L. Willham. 1995. Genetic parameter estimates for racing time by restricted maximum likelihood in the Thoroughbred horse of Japan. JABG January‐December   Pages 146-150

Menéndez Buxadera Alberto and  Marcilio Dias Silveira da Mota, 2008 Variance component estimations for race performance of thoroughbred horses in Brazil by random regression model.  Livestock Science 117  298–307

At the same time they have interpreted the suggestions of the  models strictly in formal terms when the purpose was to compare them with the current model that is made in Turkia. The same purpose of evaluation is maintained for this trait in Arabian horses and does not specify that it is in Turkia conditions. In the current version includes in the section of material and method unnecessary statistical results. The formulas of how the Legendre polynomials are estimated (l 154-163) are no longer necessary and do not contribute anything to the article . The discussion of results are inappropriate by including references to goats (l 349-350) and dairy cattle (l 387-390).

My recommendation. . The article should not be accepted.

Reviewer 2 Report

Corrections performed on the first version of the manuscript animals-1617827 are satisfactory. In my opinion, the manuscript is now suitable for publication.